# The Influence of Cecal Microbiota Transplantation on Chicken Injurious Behavior: Perspective in Human Neuropsychiatric Research

**DOI:** 10.3390/biom14081017

**Published:** 2024-08-16

**Authors:** Yuechi Fu, Heng-Wei Cheng

**Affiliations:** 1Department of Animal Sciences, Purdue University, West Lafayette, IN 47907, USA; fu263@purdue.edu; 2Livestock Behavior Research Unit, USDA-ARS, West Lafayette, IN 47907, USA

**Keywords:** social stress, aggression, injurious behavior, gut microbiota, cecal microbiota transplantation, mental disorder, chicken, human

## Abstract

Numerous studies have evidenced that neuropsychiatric disorders (mental illness and emotional disturbances) with aggression (or violence) pose a significant challenge to public health and contribute to a substantial economic burden worldwide. Especially, social disorganization (or social inequality) associated with childhood adversity has long-lasting effects on mental health, increasing the risk of developing neuropsychiatric disorders. Intestinal bacteria, functionally as an endocrine organ and a second brain, release various immunomodulators and bioactive compounds directly or indirectly regulating a host’s physiological and behavioral homeostasis. Under various social challenges, stress-induced dysbiosis increases gut permeability causes serial reactions: releasing neurotoxic compounds, leading to neuroinflammation and neuronal injury, and eventually neuropsychiatric disorders associated with aggressive, violent, or impulsive behavior in humans and various animals via a complex bidirectional communication of the microbiota–gut–brain (MGB) axis. The dysregulation of the MGB axis has also been recognized as one of the reasons for the prevalence of social stress-induced injurious behaviors (feather pecking, aggression, and cannibalistic pecking) in chickens. However, existing knowledge of preventing and treating these disorders in both humans and chickens is not well understood. In previous studies, we developed a non-mammal model in an abnormal behavioral investigation by rationalizing the effects of gut microbiota on injurious behaviors in chickens. Based on our earlier success, the perspective article outlines the possibility of reducing stress-induced injurious behaviors in chickens through modifying gut microbiota via cecal microbiota transplantation, with the potential for providing a biotherapeutic rationale for preventing injurious behaviors among individuals with mental disorders via restoring gut microbiota diversity and function.

## 1. Mental Disorder, Aggression, and Serotonin Deficiency

### 1.1. Mental Disorder, Social Stress, and Aggression

Mental disorders are a rising global prevalence influenced by multiple pathophysiological and socioeconomic factors [1]. Recent studies have revealed that mental health including emotional, psychological, and social wellness within an intergroup is associated with individuals’ (phenotypical) differences in their biological functions in response to internal and external stimulations [2,3]. Mental wellness is a critical factor influencing an individual to adapt to various challenges at all life stages from childhood to adulthood [4]. It has been defined as “*a state of well-being in which every individual realizes their own abilities, can cope with the normal stresses of life, can work productively and fruitfully, and is able to make a contribution to community*” [5]. As a hallmark of neuropsychiatric disorders, a variety of mental disorders (or disturbed emotional states) have been included in the Global Burden of Diseases, Injuries, and Risk Factors studies since 2019, such as depressive disorder, anxiety disorder, bipolar disorder, schizophrenia, autism spectrum disorder, and obsessive–compulsive disorder with overlapping mechanisms including the microbiota–gut–brain axis (MGB) and signaling events [6,7]. Recently, mental disorders have been recognized to be the leading cause of society disability with a huge economic burden globally [8]. Over one billion people, approximately 12.5% of the global population, are living with a mental disorder worldwide [9]. In the United States, this burden associated with morbidity and mortality is rising at all sociodemographic levels; one in six (16.5%) youth aged 6–17 and one in five (18.4%) adults experienced a mental illness annually [10,11]. It costs an estimated USD 5 trillion a year and is projected to increase to USD 16 trillion by 2030 globally [12]. Aggression has become one of the most common symptoms of certain mental disorders during early development [13].

Aggression, as an evolutionarily conserved behavior, is a highly complex behavior (categorized as proactive or reactive) produced by integrating signals received from the physical, physiological, and cognitive processes [14,15,16]. Normally, aggression in animals, serving as an essential component of their social behaviors to protect valuable resources, is related to survival, growth, and reproduction (appearing as an adaptive function to their living environments) [17], while under stressful conditions, aggression, as abnormal, unprovoked, or reactive behavior, is often destructive and maladaptive with long-term negative impact (causing pain, fear, excessive stress and/or injury). In particular, extreme and/or persistent forms of aggression (impulsive violence), as a symptom of certain mental disorders, have brought great attention due to causing various damage to society (as a conception of social contagion brings hurt to other persons, family, and community), resulting in significant social destruction and economic losses [18,19]. One characteristic of aggression is defined by social contagion: as a spreading behavior (such as social learning behavior and reward-seeking behavior), it can be spontaneously spread within a group, affecting interactions between individuals (i.e., establishing social rank and influencing social aggregating among members) [18]. 

A stressful environment (such as poor and/or crowded living conditions) is one of the major factors causing aggression (or violence) in a society, especially in persons with mental illness [20,21]. Stress has been defined as a state of an organism’s homeostasis being challenged, i.e., exhibiting an insufficient biophysiological response to mental, emotional, or physical challenges [22]. Stress could have a positive or negative impact on brain function based on its type as well as frequency, duration, and intensity, especially in children and adolescents (under 18 years of age) [23]. Early postnatal development is a sensitive period with extensive neuronal and behavioral plasticity (programmatic modeling), and early-life stress impairs brain programming (modulation of synaptic connection and transmission) and associated mental development, increasing the risk of developing mental illness with a long-lasting effect [24]. Although not all individuals exposed to childhood adversity develop psychopathologic disorders (exhibiting a broader phenotype under the same stress), current epidemiological research reveals that millions of children are affected by mental illness annually [25]. Numerous studies indicate that patients with aggressive (or violent) behavior are heterogenous grouping, and their violence reflects the mechanisms associated with stress phenotypes regulating the biobehavioral pattern through the biological, psychodynamic, and behavioral pathways, which is affected by the nature and quality of their social life (social disorganization) [26]. 

### 1.2. Serotonin Deficiency Theory

Various environmental, social, and biological factors are associated with the development of aggressive behavior, and stress-induced serotonin (5-HT) deficiency has become the center of aggressive research (the serotonin deficiency theory in aggression) in both patients with mental disorders and experimental animals [27,28,29], including chickens [30,31,32] (Figure 1). Serotonin with its complex functions is essential for the regulation of emotional and social behaviors, particularly for the control of aggression [33]. In the central nervous system (CNS), all components of the neural circuits involved in cognitive and emotional processes (regulating mood, anxiety, and aggressive behaviors), such as the hypothalamus (a component of the limbic system), are substantially innervated by 5-HT neurons from the dorsal and median raphe nuclei in the brainstem [34,35]. In humans, patients with psychiatric disorders, such as depressive disorder, obsessive–compulsive disorder, and anxiety disorders, have decreased brain 5-HT concentrations [36]. In addition, aggressive animals exhibit lower levels of 5-HT in the hypothalamus [37], while an experimental increase of 5-HT and/or 5-HIAA (5-Hydroxy indoleacetic acid, the primary metabolite of 5-HT) in the brain locks or retacks killing behavior in rodents [38] and fishes [39]. 5-Hydroxy indoleacetic acid has been used as a biological marker for psychiatric disorders [40]. Oral administration of 5-Hydroxytryptophan (5-HTP, an intermediate in the biosynthesis of 5-HT) shows an antidepressant-like effect in mice with depression-like behaviors via restoring gut microbiota dysbiosis [41]. In addition, increased anxiety-like behavior and impaired fear memory were found in 5-HTT−/− mice (genetically deficient in serotonin transporter (5-HTT) expression) in both familiar and unfamiliar environments compared to 5-HTT+/+ mice, which is similar to the clinical observations in humans with major depressive disorder [42]. Furthermore, altering or depleting tryptophan (TRP, the 5-HT precursor) enhances emotional processing in humans [43]. Homozygous TPH2 (Tryptophan Hydroxylase-2, the rate-limiting enzyme involved in brain 5-HT biosynthesis) knockout mice (Tph2^−/−^) exhibit higher aggressiveness compared to Tph2 wildtype mice during a president–intruder paradigm [44,45]. Monoamine oxidases (MAO, a family of enzymes that catalyze the oxidation of monoamines including 5-HT) are functionally associated with antisocial and aggressive behavior [46]. In addition, several 5-HT receptors (5-HTRs), such as 5-HT1A, 5-HT1B, 5-HT2A, 5-HT2B, and 5-HT3, are present in the neurons of the brain areas implicated in aggression control, such as the hypothalamus, hippocampus, and amygdala complex [47,48]. Genetic polymorphisms of 5-HT3, for example, are associated with impulse control disorders, while blockade of the receptor inhibits aggression [33]. Suppression of 5-HT neuron firing increases aggression in mice [49] through 5-HT1A [50] and 5-HT1B receptors [51] as well as 5-HT2B receptors [52]. However, several studies reveal that the function of 5-HT in regulating aggressive behavior is very complex, and conflicting findings against the serotonin deficiency hypothesis of aggression have been reported [53,54]. Existing knowledge of the pathophysiology of aggressive behavior is incomplete, and treatments are inadequate. There is no simplistic one-to-one relationship about 5-HT exerting an inhibitory effect on aggression. New insights into better assessing the interaction between stress and aggression as well as developing novel biotherapeutics are urgently needed. Suitable animal models are crucial in investigating the bioactive factors and underlying mechanisms involved in human aggression and mental disorders. 

## 2. Stress and Injurious Behavior in Domestic Chickens

Gene (specific genetic polymorphisms), environment (surrounding conditions), and gene–environment interaction are major risk factors for stress generation [55,56], which affects individual responses to stimulations throughout a person’s lifespan [57]. Similarly, the effects on stress response and associated health and welfare have been identified in domestic chickens [58,59]. 

Chickens were domesticated from red junglefowls 7000–10,000 years ago [60]. During domestication, chicken physiological and behavioral characteristics have been changed to adapt to the artificial rearing environments, i.e., accompanied by the availability of shelter, food, and water with less predation and diseases and different social conditions [61,62]. In particular, in recent decades, commercial chickens (both broilers and layers) have been further selected for high poultry meat or egg production to meet growing public demand [63]. During breeding, functionally integrating the signals received from the physiological and behavioral modifications as well as the artificial environments may create traits in chickens upon selection. However, recent studies have revealed that extreme selection for production traits affects other physiological and behavioral traits, declining birds’ health and welfare as a whole organism [64,65]. Generally, an animal’s productivity is correlated with its competitive ability within a flock [66,67]. Current studies have evidenced that selection for a phenotypic characteristic associated with productivity affects other traits regulating their behavioral and/or biophysiological adaptability, resulting in interspecific competition and aggression [68,69]. Aggression leads to social stress, and stress susceptibility of chickens is a major problem facing the poultry industry. These findings support our hypothesis that in chickens, as seen in humans and other species of social animals, breeding-related aggression is a highly complex social behavior. Although aggression is essential for individual chickens to establish and maintain their social status, such as through a pecking order (i.e., alpha chickens dominating other chickens), to protect valuable resources and reproduce successfully within a flock [14,70], excessive aggression can result in devastating social consequences with increased fear, stress, injury, and even cannibalism [71]. A recent study has reported that a high occurrence of feather pecking in commercial laying hens is correlated with severe plumage damage and skin injuries [72]. 

Personality differences in injurious pecking have been evidenced in chickens [73,74], which may be related to the selection for production with unequally affecting their adaptability to the rearing environments. Like the personality difference in humans, within a socioecological environment, not all individuals of chickens within a flock have an equal ability to recast their biological and behavioral characteristics in response to management-associated stressors (inter-individual differences in adaption) [18,75,76]. For example, not all chickens are (feather) peckers, while some may develop and persist in feather pecking. Generally, subordinates directly contented with a dominant individual (bully victims) within a social group intend to reduce their adaptive capacity to the rearing environments and related management practices (like the outcomes of bullying behavior in humans). Injurious behaviors are an eminent cause of chicken suffering and elevated mortality, which is a major concern in all current hen-rearing environments including both cage systems (both conventional cage and enriched cage systems) and cage-free aviary systems [77,78,79]. Until now, the conventional cage system is the most common housing facility for laying hens in the United States and most non-EU countries. In the United States, an estimated 66% of commercial eggs are derived from caged hens [80]. Commonly, laying hens are housed in groups ranging from five to nine birds per cage at a density of 64–68 or up to 86 in^2^/hen [81]. The conventional cage system may cause significant stress in chickens due to its high stocking density and a barren environment limiting hens from displaying their natural (inheritable) behaviors, such as foraging, dustbathing, exploring, perching, and nesting, resulting in a chronic state of stress [82,83]. Chickens without adapting to the rearing environment enter a “pre-pathological state” or “pathological state”, exhibiting physiological and metabolic disturbances [84] with increased abnormal aggressive behavior through the brain award system as well as the reinforcement learning pathways, which is similar to the brain award systems reported in humans [85,86]. 

## 3. Chicken Model in Psychological Disorders

### 3.1. Chicken as an Animal Model in Biological Research

Although the neural mechanisms (the core aggression circuit) underregulate response to various simulations across species, the selection of appropriate animal models is a critical factor in scientific research [87,88,89], especially in mental disorders. Compared to rodents routinely used as animal models for basic and clinical investigation, chickens (*Gallus gallus* domesticus) have served as a key animal model in almost all scientific fields, including embryology, genetics, virology, physiology, immunology, oncology, and behavior [90,91,92,93], because chickens have combined experimental advantages of both humans and rodents, avoiding the potential confounding variables such as environmental factors and genetic background [94]. A laying hen produces 250–300 eggs during her lifetime with the potential for producing numerous embryos, chicks, and chickens (next generation) with similar genetic backgrounds, but independent of maternal hormonal, metabolic, and hemodynamic influences on the neuroembryogenesis and continuous development during both prenatal and postnatal stages [95,96,97]. Chicken embryos can be easily manipulated in vivo and in vitro. In addition, eggs can be hatched in a more controllable biosecurity environment compared to delivering babies in humans and rodents. During birth in humans as well as other mammals, infants can be exposed to bacteria from numerous sources, including the birth canal, parents, hospital workers, and surrounding environment, by which bacteria can colonize in the infant’s gut with long-lasting effects [98,99,100], while in chickens, collected eggs are cleaned and sanitized immediately, stored in clean and disinfected rooms with good ventilation, then are hatched in disinfected equipment, and the hatched chicks are handled by well-trained personnel under a biosecurity environment [101]. Furthermore, as the first farm animals, chicken has its genome sequenced and analyzed, with approximately 60% genetic similarity with humans [102]. In addition, within a flock, chickens have a social hierarchy referred to as dominance rank or pecking order [103]. Like other social (group living) animals, some chickens (peckers) initiate feather pecking, then spreading throughout an entire flock (a learning or rewarding behavior). Recently, chickens have been used as an animal model in assessing the effects of genetics, environments, and genetic-environmental interactions on cognitive and social complexity as well as mental disorders and aggression [97]. Specifically, we have used two diversely selected chicken lines (aggressive vs. nonaggressive lines) as animal models in investigating the effects of modification of brain serotonergic activation on injurious behaviors [97,104,105,106,107].

### 3.2. Chickens as an Animal Model in Mental Disorder Research

Aggression in natural selection is associated with competition for survival and reproduction in social animals. Similarly, in artificial production environments, such as in the poultry industry, chickens keep a similar function of aggression during social interactions. Although there are dissimilarities between humans and chickens, the brain neural circuitries for aggression and social behavior appear to be evolutionarily conserved across the vertebrates [18,108,109]. For example, birds’ brains possesses a core “social behavioral network” that is humongous to the social behavioral network of mammals [110,111,112]. The central nuclei that are involved in cognition and emotion in avians, at least in part, are morpho- and bio-functional homologous to the mammalian nuclei [113], such as the hypothalamus [114], the hippocampus [115], the nucleus taeniae (homolog of the amygdala in mammals) [116,117], and the raphe nucleus [112,118]. These nuclei exert similar integrative functions in response to environmental stimulations. Particularly, there are similar distributions of 5-HT neurons and neurotransmitter receptors, including 5-HT receptors between birds and mammals [119]. Like the serotonergic function in humans, 5-HT [30] and selective 5-HT antagonism [120] modulate chickens’ aggression through binding to 5-HT1 and 5-HT2 receptors. Serotonin metabolism is also involved in stress-induced malfunctioned inflammatory response and related feather pecking behavior in laying hens via the MGB axis [121]. In addition, Tryptophan has long been used to attenuate aggressive behavior and control stress via regulating brain 5-HT synthesis in humans and various animals [97,122]. In chickens, lower plasma TRP concentrations have been found in feather peckers compared to their non-pecking counterparts [123,124]. Dietary supplementation of TRP (a tryptophan-enriched diet) decreases aggression in feed-restricted male chickens by centrally enhanced neuronal firing [125,126], while a tryptophan-deficient diet increases stress levels and pecking behavior in chickens [126] and increases body lesions from tail-biting and ear-biting in pigs [127]. In addition, animals fed with tryptophan-enriched diets have elevated serotonergic activity (5-HIAA/5-HT ratio, an indicator of serotonergic activity) in the hypothalamus with a decreased stress response accompanied by a significant reduction in corticosterone levels when they were exposed to social mixing stress [128,129]. Recent studies have also revealed that the function of avian stress regulatory systems (such as the hypothalamic–pituitary–adrenal (HPA) axis) is evolutionarily similar among vertebrates including humans [130,131,132]. There are similar distributions of cortisol-like compounds and their receptors (one of the stress indicators) in the same organs of both birds and mammals [133]. Taken together, birds have been used as an animal model in studying chronic stress affecting cognitive processes (emotion, fear, and memory) and related behaviors [134,135,136]. Early-life stress can cause similar long-lasting biological changes and abnormal behavior in both humans and chickens as well as other animals (Figure 2). In our previous studies, we revealed that the selection of chickens for resistance (line 6_3_) or susceptibility (line 7_2_) to Marek’s disease caused biological changes and stress responses, which are correlated with the lines’ behavioral differences. Compared to line 6_3_ birds, line 7_2_ birds exhibit greater aggression with a lower level of brain serotonergic activity [137]. The dysregulation of the serotonergic system has been implicated in a range of neuropsychiatric disorders in humans and various animals, including chickens [31,138]. These results indicate that the divergently selected inbred chicken lines may be a useful model for investigating aggression and understanding the roles of the intestine microbiota in regulating damage behaviors.

## 4. Gut Microbiota and the Gut–Brain Axis

### 4.1. Gut Microbiota in Stress Response

Numerous studies have documented that gut microbiota and their metabolites exhibit direct and indirect influences on human health and the pathogenesis of neuropsychiatric disorders [139,140,141,142]. Normally, the gut microbiota function as an endocrinological organ and a second brain with broad impacts on biological homeostasis in a host, including effects on nutrient digestion, pathogen colonization, intestinal barrier integrity, immune function, and synthesis of neuroactive factors to regulate the function of the MGB axis via multiple pathways, such as the HPA axis, in response to the internal and external stimulations [143,144]. Intestinal microbiota, especially Firmicutes and Bacteroidetes, produces antimicrobial peptides and neuroactive compounds, such as short-chain fatty acids (SCFAs) and TRP, reducing or preventing local and systemic inflammation and infection, sequentially regulating mental and behavioral health [145,146,147]. Bacteria synthesizing and metabolizing TRP (as an “essential” amino acid) within the gastrointestinal tract is essential in maintaining gut and systemic homeostasis [148,149]. Under stress, the microbiota serves as a critical social regulator and an anti-inflammatory factor that regulates stress-induced negative reactions, such as reducing impaired social behaviors and emotional expression in the hosts [150], especially during the early life co-developing of the brain, gut microbiome, and behavior [151]. Increasing studies indicate that gut microbiome influences the development of personality traits [152], regulating mood and mental health, while disturbed microbiota diversity and/or function (dysbiosis) are a significant risk factor for mental disorders [153]. Various human metagenomes (a person’s genetic composition and resident microorganisms) have been revealed among individuals; in particular, the populations of intestinal microbes (the core microbiome) vary widely with different effects on the fitness of hosts in stress–host interactions [2,3,154,155]. Differences in gut microbiota are also associated with stressful event-related resilience and susceptibility (individuals’ different responses to an activated complex, a non-specific stress regulating system) for the development of neuropsychiatric disorders. It is one of the reasons that some individuals are more sensitive to stress stimuli than others even under the same or similar living conditions [3]. 

The surrounding (living) environment is overflowing with various social interactions, such as social engagements (fight-or-flight) and intergroup alterations (stability–instability), which function as biopsychological (emotional and mental overstimulation), physical (fitness), and environmental (social conditions) stressors in humans and various animals [156]. Gut microbiota, as a stress sensor, is sensitive and reactive to various stimulations, resulting in serial disorders of the multiple signaling systems pathways of the MGB axis, including the neurotransmitter, endocrine, immune, and various biochemical pathways [141], consequently leading to the destruction of gut microbial profile (diversity, composition, or both), increasing permeability, releasing pathogenic bacteria and bacterial metabolites, suppressing local and systemic immunity, increasing the permeability of the blood–brain barrier (BBB), and causing a low-chronic neuroinflammation with the activated HPA axis. Subsequently, these changes alter brain gene expression and neurotransmitter synthesis, leading to mental disorders with abnormal behaviors [157,158,159]. Similarly, stress-induced injurious behaviors in laying hens have been recognized as the consequence of dysregulation of the MGB axis [137,160]. The MGB axis in birds, similar to it in humans, is a bidirectional biochemical signaling network between the microbiota and the brain, by which gut bacterial metabolites and synthesized neuroactive factors influence the functions of the brain to effectively affect mood, cognition, and mental health [161,162,163].

Increasing clinical and preclinical research shows there are significant associations between the alterations in the abundance of specific microbial taxa (gut microbial composition and diversity) and mental disorders, including depression, generalized anxiety disorder, autism spectrum disorder (ASD), and schizophrenia. Butler (2023) reported that the gut microbial structure (i.e., beta diversity) was different between patients with social anxiety disorder (SAD) patients and controls, i.e., healthy controls had enriched *Parasuterella*, while the relative abundance of the genera *Anaeromassillibacillus* and *Gordonibacter* was elevated in SAD patients [164]. Following a systematic review, Korteniemi et al. (2023) indicated a correlation between ASD symptoms and the composition of gut microbiota [165]. Higher levels of Proteobacteria, Actinobacteria, and *Sutterella* were found in ASD children. In a cross-sectional study, compared to healthy controls, specific genus-level microbial panels of *Ruminococcus*, *UCG005*, *Clostridium_sensu_stricto_1*, and *Bifidobacterium* were increased in schizophrenia patients [166]. Distinct characteristics and diversity of gut microbiota, including *Flavonifractor plautii*, *Ruminococcus gnavus*, and *Bifidobacterium* species, were enriched in individuals who developed adverse post-traumatic neuropsychiatric sequelae [167]. Buey et al. (2023) also reported that the growth in SCFA-producing beneficial bacteria regulates the intestinal serotonergic system via modification of the function, synthesis, and expression of 5-HT, 5-HTT, and 5-HTRs [168]. In addition, declined fermentative taxa have been associated with different psychiatric disorders, resulting from a declined production of SCFAs and an increased synthesis of pro-inflammatory factors [169]. Fermented foods containing beneficial microbes, microbial metabolites, and other bioactive compounds have become one of the novel therapeutic approaches targeting neuropsychiatric and neurodegenerative disorders [170,171]. 

The gut microbiota influencing the development of neuropsychiatric disorders is further revealed in experimental animals. The animals raised in a germ-free (GF) environment expressing an enhanced HPA response to psychological stressors could be reduced and normalized with certain bacteria as probiotics, such as *Bifidobacterium infantis* [172] and *Bacillus licheniformis* [173]. A blunted HPA response has been found in probiotic-treated healthy female subjects [174]. In addition, antibiotics, as antibacterial agents, are commonly used in the treatment of inflammatory bowel disease [175]. However, as side effects, animals with antibiotic-perturbated microbiota composition (i.e., depleted anti-inflammatory butyrate-producing bacteria and enhanced pro-inflammatory bacteria) [176] experience a negative impact on their health due to disrupted gut microbial diversity leading to gut dysbiosis and related cognitive, emotional, and behavioral changes [177,178]. In the study, antibiotic-caused microbiota deletion in animals leads to increased lipopolysaccharide levels, neuroinflammation, and behavioral changes [179]. Shoji et al. (2023) also reported that compared to 5-HTT+/+ mice, altered gut microbiota abundances were found in 5-HTT−/− mice with reduced abundance of *Allobaculum*, *Bifidobacterium*, *Clostridium sensu stricto*, and *Turicibacter*, which is paralleled with increased depression-related behavior, impaired fear memory, and altered social behavior [42]. Those behaviors can be reduced by SCFA-producing bacteria, such as lactic acid-producing bacteria, via increasing the functions of both the gut and blood–brain barrier and reducing gut and brain inflammation [3]. Hou et al. (2023) and Zhou et al. (2023) reported that dysbiosis-associated disruption in TRP catabolism contributes to neuroinflammation and psychiatric disorders [180,181]. These results further indicate that the maintenance of gut microbial balance and function is essential for humans and animals (including chickens) to keep their optimal biophysiological and behavioral functions in growth, reproduction, and welfare. Targeting the intestinal microbiota to restore its balance using prebiotics and probiotics as well as plant supplements, fermented foods, and fecal microbiome transplant (FMT) has been recognized as a novel strategy for patients with emotional and mental disorders [182,183].

There is increasing research into microbial-based psychopharmaceuticals. A novel class of probiotic organisms, such as the species from the genera *Lactobacillus*, *Bifidobacterium*, and *Bacillus*, have been used as psychobiotics (also named the next-generation probiotics) to deliver mental health benefits to alleviate neuropsychiatric symptoms via modulating host’s gut microbiota and the function of the MGB axis [184,185,186]. A systematic review of randomized and controlled clinical and preclinical trials revealed that the psychobiotic effect of different strains of *Lactiplantibacillus plantarum* as well as their metabolic substances play a vital role in treating neuropsychiatric disorders, alleviating a series of symptoms of disorders, such as cognitive function, memory capability, anxiety status, emotional stability, and hyperactivity, by restoring the gut microbiota diversity, reestablishing the integrity of the intestinal barrier, regulating the activity of the HPA axis, and modulating gut and brain immune functions [187]. In addition, since 2016, in preclinical and human trials, *Bifidobacterium* and *Lactobacillus genera* (*B. longum*, *B. breve*, *B. infantis*, *L. helveticus*, *L. rhamnosus*, *L. plantarum*, and *L. casei*), as natural alternatives to traditional therapeutic drugs, have been used for preventing and treating patients with mental disorders, such as depression, anxiety, bipolar, schizophrenia, temper tantrums, impulsivity, and compulsivity, as well as neurodegeneration diseases associated with the dysregulation of the MGB [188,189,190,191]. For example, compared to placebo, probiotic *Bifidobacterium longum* reduced the stress negative effects in patients, which is correlated with a reduction in anxiety, depression, and the cortisol awakening response [191]. Compared to controls, the enriched *Lactobacillus*, *Faecalibacterium*, and *Ruminococcus* significantly improves depressive symptoms and cognitive impairments in bipolar patients [192]. Chan (2023) reported that an E3 multi-strain probiotic improved depressive anxious symptoms in patients with notable differences in beta diversity, i.e., the relative abundance of *Bifidobacterium bifidum*, *Lactobacillus acidophilus*, *Lactobacillus helveticus*, and *Lactobacillus plantarum* as well as increased Firmicutes/Bacteroidetes (F/B) ratio [193]. The probiotic *Lacticaseibacillus rhamnosus IDCC3201 (L3201)* [194] and *Lactobacillus reuteri ATG-F4* [195] also improve the anxious and depressive-like behavior, accompanied by biochemical changes in TRP metabolism in unpredictable chronic mild stressed mice, respectively. As a psychobiotic, *Akkermansia muciniphila* also shows effects on brain functions, reducing the symptoms in various neurological and psychiatric disorders with the potential therapeutic application [196,197]. Overall, the current evidence suggests that (1) The gut microbiota and their metabolites regulate hosts’ mental state and behavioral development; (2) stress-induced dysbiosis is involved in developing neuropsychiatric disorders; (3) specific strains of probiotics (commensal bacteria) can reduce psychiatric symptoms by regulating the mechanisms including neurotransmitter synthesis, modulation of inflammatory cytokines, and/or stress response affecting the release of stress hormones; and (4) psychobiotics may offer therapeutic benefits for managing neuropsychiatric disorders (Table 1). The evidence reveals that the use of psychobiotics to treat neuropsychiatric disorders could impact future clinical practice, as current treatments for these diseases have limited efficacy due to their complex pathogenesis.

Recent studies have been increasingly focusing on another scientific area, fecal microbiota transplantation (FMT), i.e., transferring stool from a healthy donor, as a novel biotherapy for psychiatric and psychological disorders. The gut microbiota is the only “organ”, i.e., a core microbiome, shared among all healthy adults, although there is variation over time and post-stimuli across populations [198,199]. Both clinical and preclinical studies evidence that gut microbiota can transfer behavioral features from donors to recipients through colonization and/or synthesis of neuroactive compounds, such as SCFAs, glutamate (Glu), γ-aminobutyric acid (GABA), dopamine (DA), norepinephrine (NE), 5-HT, and histamine [142,200]. Fecal microbiota transplantation is a clinical procedure with benefits against gut dysbiosis-associated brain disorders by promoting growth in SCFA-producing bacteria, enhancing the gut barrier, reducing the gut inflammatory response, and maintaining BBB integrity. 

Fecal microbiota transplantation via modulating gut microbiota, recently, has become a novel method for treating gastrointestinal disorders, such as inflammatory bowel syndrome and recurrent *C. difficile* infection (CDI) [199,201] as well as neuropsychiatric disorders and neurodegenerative disorders [202]. Stool contains various microorganisms and their metabolites, and FMT may be a rapid and effective method to reestablish the intestinal microbiota and metabolic profiles in humans and animals [203,204]. For example, transplanting microbiota from chronically stressed rats induced stress on control animals, and such stress could be reversed by transferring microbiota from unstressed animals [205,206]. In addition, FMT has been used to alter the gut composition by reducing *Escherichia–Shigella* in patients with generalized anxiety disorder via regulating the MGB axis [207]. Based on these outcomes, the FDA has approved the first FMT agent, Rebyota, for treating recurrent and refractory CDI [208]. Studies in CDI patients revealed that the gut microbiota diversity is increased following FMT, which is critical for resisting pathogen colonization. Clinically, a single dose can have long-lasting effects [209,210]. In addition, the effects of restoring microbiome composition on impulsive and violent behavior have become an important target for the treatment of stress-related mental disorders. However, the pathogenesis of mental disorders involves complex interactions between gut microbiome dysbiosis, immune hyperactivity, and neurotransmitter disorders, which is poorly understood. Chicken could be a good model for mental disorder research as it has a similar effect of modulating gut microbial composition in response to stress challenges that present in laying hens housed in artificial rearing environments. 

**Table 1 biomolecules-14-01017-t001:** Examples of clinical and preclinical research on gut microbiota and neuropsychiatric disorders.

Model/Disorder	Findings	References
5-HTT+/− mice	5-HTT+/− mice showed slightly reduced locomotor activity and impaired social behavior compared to 5-HTT+/+ mice. Also, 5-HTT−/− mice had decreased abundance of *Allobaculum*, *Bifidobacterium*, *Clostridium sensu stricto*, and *Turicibacter*, compared to 5-HTT+/+ mice.	[42]
Social anxiety disorder(SAD) patients	Beta-diversity was different between the SAD and control groups. Also, several taxonomic differences were found, i.e., genera *Anaeromassillibacillus* and *Gordonibacter* were elevated in SAD, while *Parasuterella* was enriched in healthy controls. At a species level, *Anaeromassilibacillus* sp. *An250* was found to be more abundant in SAD patients, while *Parasutterella excrementihominis* was higher in controls.	[164]
Autism spectrum disorder (ASD) patients	ASD children’s gut microbiota had higher abundance of *Proteobacteria*, *Actinobacteria*, and *Sutterella* compared to controls.	[165]
Schizophrenia patients	Specific genus-level microbial panels of *Ruminococcus*, *UCG005*, *Clostridium_sensu_stricto_1*, and *Bifidobacterium* were increased in schizophrenia patients.	[166]
Adverse post-traumatic neuropsychiatric sequelae (APNS) patients	Microbial species, including *Flavonifractor plautii*, *Ruminococcus gnavus* and, *Bifidobacterium* species were found to be important in predicting worse APNS outcomes from microbial abundance data.	[167]
Chronic unpredictable mild stress (CUMS) model rats	*B. licheniformis* reduced depressive-like and anxiety-like behaviors in rats during the CUMS process. Also, *B. licheniformis* changed the gut microbiota composition; increased the short-chain fatty acids (SCFAs) in the colon; decreased kynurenine, norepinephrine, and glutamate levels; and increased the tryptophan, dopamine, epinephrine, and γ-aminobutyric acid (GABA) in the brain. Meanwhile, *Parabacteroides*, *Anaerostipes*, *Ruminococcus-2*, and *Blautia* showed significant correlation with neurotransmitters and SCFAs.	[173]
Healthy female subjects followed a working memory task	The relative abundance of eight genera in the probiotics group was higher (uncorrected) relative to the placebo group: *Butyricimonas, Parabacteroides, Alistipes, Christensenellaceae_R-7_group*, *Family_XIII_AD3011_group*, *Ruminococcaceae_UCG-003*, *Ruminococcaceae_UCG-005*, and *Ruminococcaceae_UCG-010*. Also, the probiotic-induced change in genus *Ruminococcaceae_UCG-003* was significantly associated with probiotics’ effect on stress-induced working memory changes compared to the placebo group.	[174]
Common marmosets (*Callithrix jacchus*) with antibiotic cocktail	Increase in gut *Fusobacterium spp*. post-antibiotic administration with significant changes in concentrations of several gut metabolites that are either neurotransmitters (e.g., GABA and 5-HT) or the moderators of gut–brain axis communication in rodent models (e.g., short-chain fatty acids and bile acids). Also, antibiotic-administered marmosets exhibited increased affiliative behavior and sociability, which might be a coping mechanism in response to gut dysbiosis-induced stress.	[177]
Meta-analysis of 48 rodent studies	The increased abundance in gut Proteobacteria showed statistically significant association with an increase in anxiety.	[178]
49 eligible volunteers	Compared to placebo, probiotic *Bifidobacterium longum* reduced the stress negative effects in patients, correlated with a reduction in anxiety, depression, and the cortisol awakening response.	[191]
Subjects with sleep disturbance and mood symptoms	An E3 multi-strain probiotic improved depressive anxious symptoms in patients with notable differences in beta diversity, i.e., the relative abundance of *Bifidobacterium bifidum*, *Lactobacillus acidophilus*, *Lactobacillus helveticus*, and *Lactobacillus plantarum* as well as increased Firmicutes/Bacteroidetes (F/B) ratio.	[193]
Chronic Stress Mouse Model	Probiotic *Lacticaseibacillus rhamnosus IDCC3201 (L3201)* and *Lactobacillus reuteri ATG-F4* improved anxiety-like and depressive-like behaviors, accompanied by biochemical changes in tryptophan metabolism of unpredictable chronic mild stressed mice, respectively.	[194,195]
Generalized anxiety disorder (GAD) patients: meta-analysis	Levels of specific enteric bacteria (*Escherichia–Shigella*) are increased in GAD patients; 71% of reviewed studies showed decreased levels of *Escherichia–Shigella* post-FMT compared to pre-FMT, suggesting FMT could be as a treatment modality for GAD.	[207]

### 4.2. Gut Microbiota in Chickens

The gut of chickens and humans shares similar dominant bacterial phyla, including Firmicutes, Bacteroidetes, Actinobacteria, and Proteobacteria [211,212], and the first two comprise approximately 90% of the gut microbiota [213]. In addition, chicken gut microbiome, like it in mammals, plays a critical role in poultry health [214,215,216]. In chickens, stress damages intestinal bacterial composition and the bilateral connection of the MGB axis, resulting in the destruction of pathophysiological homeostasis and behavioral exhibition [217], especially during early post-incubated development. In humans and other mammals, the early-life stage is a critical window for the co-development of both the microbiome and the host, and gut microbiome plays a critical role in maintaining homeostatic regulation and influencing host behavioral appearances [99,218]. It has been evidenced that transient changes in the microbiome during this sensitive stage of life can cause long-lasting effects [78,99,219]. Similarly, changing brain developmental programming during the prenatal and early postnatal periods are involved in the development of injurious behaviors in laying hens [138] and other farm animals [220]. Recent studies have indicated that exhibition of injurious behavior in chickens is influenced by dysregulation of the gut microbiome, consequently affecting neurotransmitter and immune homeostasis [221,222,223]. In addition, the populations of microbes can vary widely between individuals. Laying hens showing high or low feather pecking have different gut microbial compositions [224] and relative metabolite profiles [225]. Therefore, modification of the gut microbiome may represent a novel strategy for preventing stress-induced mental and emotional disorders in humans and injurious behaviors in laying hens. Current studies have revealed mammals and birds share the functions of the gut microbiota on the host behavior [226], and probiotics show a similar function in restoring gut microbiota composition and function [227]. Dietary supplements of several probiotics have been used with therapeutic targets for restoring the gut microbiota and 5-HT metabolism for reducing abnormal behaviors, such as *Bacillus amyloliquefaciens* reducing distress calls and aggressive behavior in turkey poults [228], and *Lactobacillus rhamnosus* [229] and *Bacillus subtilis* decreasing stress-induced feather pecking in adult hens [230]. However, the evidence for the beneficial effects of probiotics is mixed with several weaknesses including transient beneficial effects and required continuous administration over time due to the colonization and resistant interactions from the host’s resident microbial populations [231]. Using live commensals, like FMT, may be more effective than probiotics.

### 4.3. Cecal Microbiota Transplantation and Injurious Behavior in Chickens

The avian cecum has a greater biological role in maintaining biophysiological homeostasis than the cecum in most mammals [232,233]. Anatomical and microbiological analyses indicate that along the chicken gastrointestinal tract, the cecum has the greatest bacterial biodiversity (bacterial diversity, richness, and species composition) [234,235] with complex motility. It functions as a two-component system pushing contents in two directions: the cloaca, excreting as cecal drop, and the ileum, providing bacteria (for bacterial proliferation and colonization) involved in developing and maintaining gut microbiota balance [236,237] and determining colonization resistance against invading pathogens [238]. It has been previously reported that different chicken lines have different cecal microbiota compositions and functions in response to environmental stressors (chronic physical or social environment with negative effects, such as heat stress) [239] and experimental immune challenges (such as experimentally infected with *Salmonella*) [240]. Laying hens showing high or low feather pecking had distinct gut microbiota composition [221] with different intestinal and systemic metabolite profiles [225]. 

Compared to fecal samples, the cecal content has higher levels of DNA replicative viability [241]. In addition, a fecal sample is not reliable in mapping and monitoring the changes in cecal content in response to stress challenges. Cecal microbiota transplantation (CMT) as well as several probiotics, such as *Lactobacillus rhamnosus* [229] and *Bacillus subtilis* [230], have been used in reducing or inhibiting feather pecking in chickens, as feather pecking has been considered to be a stress-induced mental disorder, and it is comparable to human psychopathological disorders [124]. Stress-induced gut dysbiosis increases gut permeability and low chronic inflammation are common traits of these disorders in both humans and chickens. The development and maintenance of gut microbiota may have considerable influence in host social interactions, such as the social rank (i.e., pecking order in chickens, a social hierarchy) of individual animals [242]. For laying hens, feather pecking may therefore be a phenotypic behavioral consequence of imbalanced gut microbiota composition and the dysregulation of the MGB axis [229]. Supporting the hypothesis, birds performing more feather pecking have distinct microbiota profiles compared to their non-pecking counterparts [221]. In addition, microbiota differs between the selected lines exhibiting distinct phenotypes [137,243]. Cecal microbiota transplantation may offer a novel tool for targeting abnormal behaviors in both chickens and humans.

Major colonization of gut microbiota occurs during delivery in newborns and immediately after hatching in chicks. In addition, administration of probiotics before or during hatching (such as in ovo administration) can introduce beneficial changes in gut microbiota and intestinal development in chickens [96,244,245,246]. Similarly, CMT in day-old chicks has protective effects on stress-induced pathophysiological and behavioral changes. For example, broiler chicks at day 1 (recipients) that orally receive cecal contents from hens (donors) have similar bacterial microbiota profiles compared to the donors [247,248]. In addition, fearfulness was reduced in the broiler recipients after transferred fecal content collected from healthy aged broilers (donors at 52 week of age) [249]. Those results indicate that the prenatal or early postnatal periods are vital windows for colonization of gut microbiota in both birds and animals. Modification of gut microbiome during early life profoundly influences the development of the gut–brain neural circuits with a long-term impact on reacting stressful episode challenges. However, improvement in gut microbial composition in chickens via CMT-induced intestinal microbial modulation through an early-life program has not been well established, and inconsistent results have been reported across studies. In addition, early-life homologous (within a line) microbiota transplantation had limited effects on microbiota composition, stress response, and feather pecking in recipients [250]. This may be dependent on the multiple factors such as transplanted dose and duration, delivery method, and donor’s age and health, affecting the transferred bacteria to be survived and colonized in the gut of the recipients. It is also elusive how feather pecking has arisen as a result of dysregulated communication between the gut microbiota and the brain. Some studies have reported that when variations, such as breed and diet, are created, no differences were found in the cecal microbial community diversity, and the major functional characteristics of the gut microbiota were unaltered [251]. Furthermore, a recent study reported that gut microbial composition (the digesta and mucosa of the ileum and cecum) and predicted functions are not associated with feather pecking and antagonistic behavior in laying hens [252]. Taken together, it is critical to further identify the potential bio-functions of cecal microbiota and their metabolites in controlling injurious behaviors in laying hens. We hypothesized that the developed divergently selected lines (6_3_ and 7_2_) with opposite behaviors (non-aggressive vs. aggressive lines) could be a useful model for abnormal behavioral investigation. Exploring the similarities and differences in the gut microbial composition between the two selected lines will lead to the development of innovative therapeutic interventions to enhance the health and welfare of both humans and chickens. To test the hypothesis, we designed a multi-year study to examine early-life CMT and aggressive behaviors in laying hens [137,253]. In the study, the cecal contents of line 6_3_ (gentle) and line 7_2_ (aggressive) (donors) were orally transferred to male chicks of a commercial line Dekalb XL (recipients). The most important outcome is that early-life CMT from the two divergently selected lines reduces stress-induced aggression in recipients by regulating the cecal microbiota composition, hypothalamic serotonergic activity, and stress reactions via the gut–brain axis. The findings confirm our working hypothesis that transferring cecal contents at an early age has line-specific effects on recipients, with the potential to induce long-term effects on regulating the behavioral expression of chickens. 

## 5. Conclusions

Current research has revealed that gut microbiota plays a critical role in early brain programming and later stress response in both humans and animals including chickens. Like an endocrine organ and a second brain, gut microbiota reacts to various internal and external stimuli, linking to the pathogenesis of stress effects, consequently influencing brain function in the regulation of host health and behavior through the MGB axis. This perspective article outlines the possibility of using chickens as an animal model in investigating stress-induced injurious behaviors in both chickens and humans through modifying gut microbiota. Our findings provide new insights into further understanding the cellular mechanisms of gut microbiota in regulating stress-induced abnormal behavior and provide a therapeutic strategy for reducing the risk of injurious behaviors in chickens as well as violence in humans. The outcomes could provide implications for developing novel therapeutics, psychobiotics, and psychobiotic compounds, impacting neuropsychiatric clinical practice. 

## Figures and Tables

**Figure 1 biomolecules-14-01017-f001:**
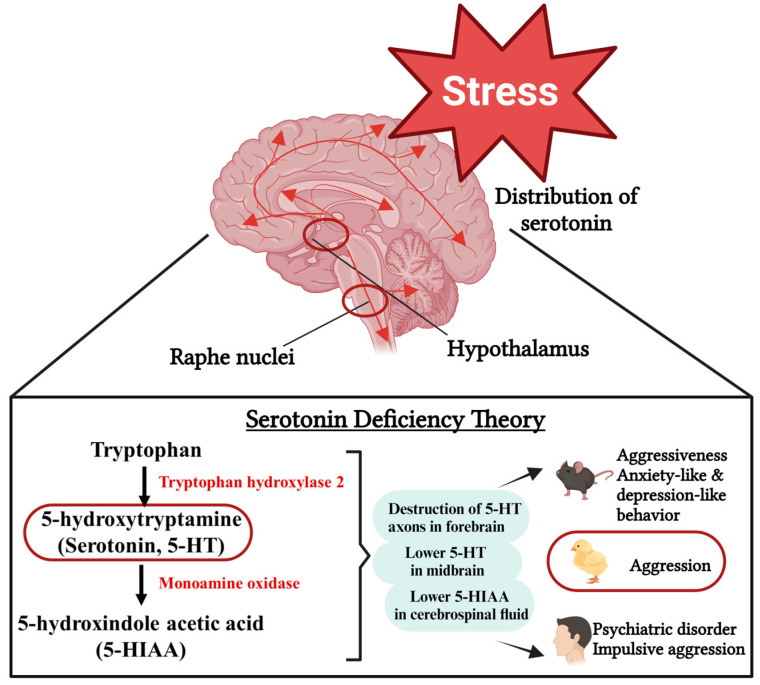
Stress-induced serotonin deficiency theory (created with BioRender.com.).

**Figure 2 biomolecules-14-01017-f002:**
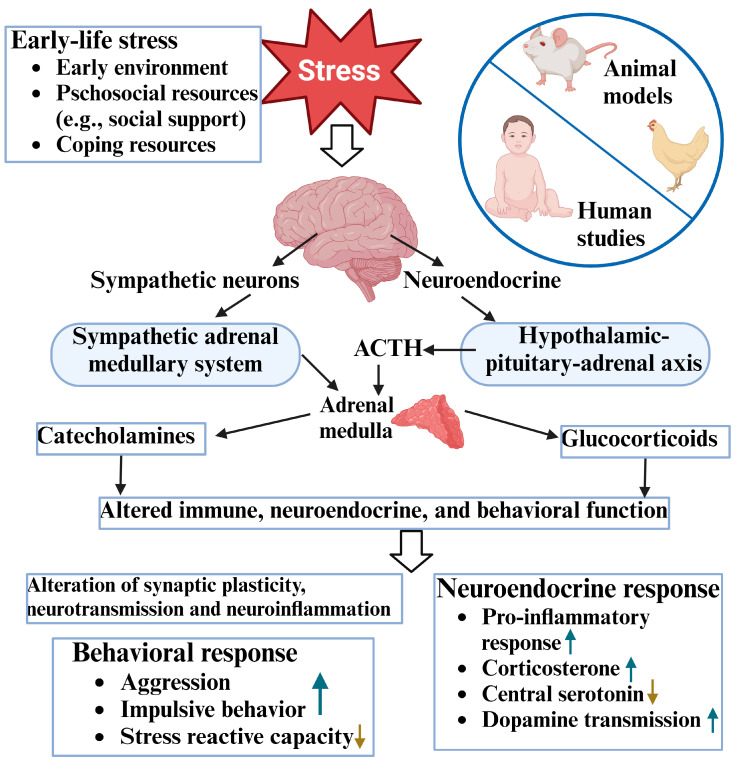
The potential mechanisms of early-life stress cause long-lasting mental disorders in humans and experimental animals, including chickens (created with BioRender.com.).

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
