# Peer review of "The Influence of Cecal Microbiota Transplantation on Chicken Injurious Behavior: Perspective in Human Neuropsychiatric Research"

_biomolecules, 2024, doi:10.3390/biom14081017_

Round 1

Reviewer 1 Report

Comments and Suggestions for Authors

Thise review is a very interesting and complete research about the use of chicken models to analyse human mental trastorns derivate of microbiota changes.

Fu and Cheng's manuscript is presented in the journal as a review. However, in the abstract it seems that original results are included (this is later clarified in the full manuscript) it would be important to add in line 27 in a previous study.

The division of the review into sections is very appropriate and provides the necessary data to corroborate the hypothesis proposed: studies of cecal microbiota transplantation in chicken can be used as a model to analyze disturbances in human behavior. The bibliography is very abundant, it is up-to-date and the authors analyze it in depth. The conclusions emerge from the analysis of the bibliography and the background of the group presented previously.

I only consider that it would be necessary to incorporate graphs and tables that summarize the different sections. For example, it would be important to construct graphs that summarize sections 1.2, 3.2. and 4.2 and from a table in section 4.1. A graph could also be incorporated that explains the experiment that the authors previously carried out and that is described in the section. 4.4.

Author Response

Response to Reviewer #1

We thank the reviewer for the complimentary comments and suggestions, which have helped us to improve our manuscript.

1. Thise review is a very interesting and complete research about the use of chicken models to analyse human mental trastorns derivate of microbiota changes.

Thank you for the comments.

2. Fu and Cheng's manuscript is presented in the journal as a review. However, in the abstract it seems that original results are included (this is later clarified in the full manuscript) it would be important to add in line 27 in a previous study.

Done as suggested (lines 50-54).

3. The division of the review into sections is very appropriate and provides the necessary data to corroborate the hypothesis proposed: studies of cecal microbiota transplantation in chicken can be used as a model to analyze disturbances in human behavior. The bibliography is very abundant, it is up-to-date and the authors analyze it in depth. The conclusions emerge from the analysis of the bibliography and the background of the group presented previously.

Thanks for the encouragement.

4. I only consider that it would be necessary to incorporate graphs and tables that summarize the different sections. For example, it would be important to construct graphs that summarize sections 1.2, 3.2. and 4.2 and from a table in section 4.1. A graph could also be incorporated that explains the experiment that the authors previously carried out and that is described in the section. 4.

As suggested, Figures 1 to 3 (lines 183, 306, 632) and Table 1 (line 1417) have been added.

Reviewer 2 Report

Comments and Suggestions for Authors

1. The study tries to make connections between the effects of cecal microbiota transplantation in chickens and possible human applications. However, without much proof or more bridge research in mammalian models, extrapolating from chicken models to human neuropsychiatric diseases would be unduly ambitious.

2. Although the methodology explains the process of gathering and transferring the contents of the cecum from various chicken lines, it might not provide adequate details about the controls or other experimental parameters, like sample sizes, which are essential for evaluating the accuracy and repeatability of the results.

3. Please validate the study's conclusions with precise statistical procedures used to analyze the data, which is not done in this review. Understanding the significance of the observed variations between experimental groups is aided by the inclusion of comprehensive statistical analysis

4. Although the study focus on serotonergic activity and behavioral results, a wider range of physiological and molecular tests could be useful to bolster the connection between behavioral changes and microbiota modifications.

5. The study's drawbacks, such as the possibility of variation in the microbiota composition of individual hens in the same line and outside influences that might affect the outcomes, might not be adequately covered in the report.

6. Graphical abstract need to change to highlight content of the paper

7. add the diagrams for section, 2 (Stress and Injurious Behavior in Domestic Chickens) 3 (Chicken Model in Psychological Disorders) and 4 (4. Gut Microbiota and the Gut-Brain Axis)

Comments on the Quality of English Language

Minor changes required on english

Author Response

Response to Reviewer #2

We thank the reviewer for the complimentary comments and suggestions, which have helped us to improve our manuscript

Comment 1: The study tries to make connections between the effects of cecal microbiota transplantation in chickens and possible human applications. However, without much proof or more bridge research in mammalian models, extrapolating from chicken models to human neuropsychiatric diseases would be unduly ambitious.

 Response 1: Gut microbiota transplantation has become a critical method used in multiple scientific fields, including neuropsychiatric disorders in humans and various experimental animals including chickens. Searching a single database, PubMed, within recent 10 years:

In rodents (a mammalian model for human investigation): using the keyword “Gut microbiota transplantation in rodents” received 2,224 outcomes; “fecal microbiota transplantation in rodents” received 1,529 outcomes, and “cecal microbiota transplantation in rodents” received 144 outcomes.

In humans:” gut microbiota transplantation in humans” received 4,579 outcomes, “fecal microbiota transplantation in humans” received 4,031 outcomes, and “cecal microbiota transplantation in humans” received 84 outcomes.

In birds (including chickens): “gut microbiota transplantation in birds” received 54 results, “fecal microbiota transplantation in birds” received 33 outcomes, and “cecal microbiota transplantation in birds” received 33 outcomes. 

Birds (including chickens), as an animal model, have been broadly used in various human behavioral investigations, and especially, in the relationship between behavior and gut microbiome in humans (Hammack and May, 2015; Moorman and Nicol, 2015; Monsoro-Burg and Levin, 2018; Kato et al., 2020; Flores-Santin and Burggren, 2021; Lyte et al., 2021; Jadhav et al., 2022).

References:

Flores-Santin J, Burggren WW. Beyond the Chicken: Alternative Avian Models for Developmental Physiological Research. Front Physiol. 12:712633.

Hammack SE, May V. 2015. Pituitary adenylate cyclase activating polypeptide in stress-related disorders: data convergence from animal and human studies. Biol Psychiatry. 78(3):167-77.

Jadhav VV, Han J, Fasina Y, Harrison SH. 2022. Connecting gut microbiomes and short chain fatty acids with the serotonergic system and behavior in Gallus gallus and other avian species. Front Physiol. 13:1035538.

Kato TA, Shinfuku N, Tateno M. 2020. Internet society, internet addiction, and pathological social withdrawal: the chicken and egg dilemma for internet addiction and hikikomori. Curr Opin Psychiatry. 33(3):264-270.

Lyte JM, Keane J, Eckenberger J, Anthony N, Shrestha S, Marasini D, Daniels KM, Caputi V, Donoghue AM, Lyte M. 2021. Japanese quail (Coturnix japonica) as a novel model to study the relationship between the avian microbiome and microbial endocrinology-based host-microbe interactions. Microbiome. 9(1):38.

Monsoro-Burq AH, Levin M. 2018. Avian models and the study of invariant asymmetry: how the chicken and the egg taught us to tell right from left. Int J Dev Biol. 62(1-2-3):63-77.

Moorman S, Nicol AU. 2015. Memory-related brain lateralisation in birds and humans. Neurosci Biobehav Rev. 50:86-102.

Comment 2: Although the methodology explains the process of gathering and transferring the contents of the cecum from various chicken lines, it might not provide adequate details about the controls or other experimental parameters, like sample sizes, which are essential for evaluating the accuracy and repeatability of the results.

Response 2: This is a review article; the detailed information has been published in citied articles (Fu et al., 2022, 2023; Hu et al., 2022). However, as suggested, some information has been provided in Figure 3, experimental design and section 4.4.

References

Fu, Y., Jiaying Hu, Marisa Erasmus, Timothy Johnson, and H.W. Cheng. 2022. Effects of early-life cecal microbiota transplantation from divergently selected inbred chicken lines on growth, gut serotonin, and immune parameters in recipient chickens. Poult. Sci. 101:101925.

Fu, Y., Timothy A. Johnson, Jiaying Hu, Marisa A. Erasmus, Huanmin Zhang, Heng-wei Cheng. 2023. Cecal microbiota transplantation: Unique influence of cecal microbiota from two divergently selected inbred donor lines on cecal microbial profile, serotonergic activity, and aggressive behavior of recipient chickens. JASB. 14:66,

Hu, J.Y., T.A. Johnson, H.M. Zhang, H.W. Cheng. 2022. The microbiota-gut-brain axis: The gut microbiota modulates conspecific aggression in diversely selected laying hens. Microorganisms. 10: 1081.

Comment 3: Please validate the study's conclusions with precise statistical procedures used to analyze the data, which is not done in this review. Understanding the significance of the observed variations between experimental groups is aided by the inclusion of comprehensive statistical analysis

Response 3: The detailed information about the statistical analysis has been published previously but added in section 4.4 (lines 602-608) for the convenience (please see the reference articles above).

Comment 4: Although the study focus on serotonergic activity and behavioral results, a wider range of physiological and molecular tests could be useful to bolster the connection between behavioral changes and microbiota modifications.

Response 4: Thanks for the reviewer to point it out. There are several neuromodulators regulate aggression in humans and animals, such as reduced serotonin activity; enhanced dopamine and norepinephrine concentrations causing the orbitofrontal/cingulate cortex processing signaling inefficiency; and reduced GABA but increased glutamate and acetylcholine levels associated with the hyperactivity of the limbic system (Miczek et al., 2002; Siever, 2008). In addition, gonadal hormones (primarily testosterone) are also involved in aggression via modulating relevant neural circuits (Soma et al., 2008). However, brain serotonin deficiency is the central for aggression research (Tricklebank and Petrinovic, 2019; Sofi et al., 2021; Salvan et al., 2022). Broadly discussing all factors involved in aggression is not the aim of this article. In this article, the change of serotonergic system is used as an example for the comparison between aggression in neuropsychiatric patients and in chickens, by which it reveals that the selected chicken lines could be used as an avian model for human neuropsychiatric disorder research.

References

Miczek, K.A., Fish, EW., de Bold JF., and de Almeida, RMM. 2002. Social and neural determinants of aggressive behavior: pharmacotherapeutic targets at serotonin, dopamine and g-aminobutyric acid systems. Psychopharmacology, 163:434–458

Salvan, P., Fonseca, M., Winkler, A.M. Antoine Beauchamp, Jason P. Lerch & Heidi Johansen-Berg 2023. Serotonin regulation of behavior via large-scale neuromodulation of serotonin receptor networks. Nat Neurosci 26: 53–63.

Siever LJ. 2008. Neurobiology of aggression and violence. Am J Psychiatry. 165(4):429-42.

Sofi da Cunha-Bang, Gitte Moos Knudsen. 2021. The Modulatory Role of Serotonin on Human Impulsive Aggression. Biological Psychiatry, 90: 447-457,

Soma, K.K., Melissa-Ann L. Scotti, Amy E.M. Newman, Thierry D. Charlier, Gregory E. Demas. 2008. Novel mechanisms for neuroendocrine regulation of aggression,Frontiers in Neuroendocrinology, 29: 476-489.

Tricklebank, M.D., Marija M. Petrinovic. 2019. Chapter Nine - Serotonin and aggression. Editor(s): Mark D. Tricklebank, Eileen Daly, The Serotonin System, Academic Press, Pages 155-180.

Comment 5: The study's drawbacks, such as the possibility of variation in the microbiota composition of individual hens in the same line and outside influences that might affect the outcomes, might not be adequately covered in the report.

Response 5: We agree with the comment that “the possibility of variation in the microbiota composition of individual hens in the same line and outside influences that might affect the outcomes, might not be adequately covered in the report.”.

Concerning with this issue and avoiding its effects during conducting the studies, the cecal samples from 10 hens per line were pooled before oral administration. This procedure is similar to fecal microbiota transplantation which is commonly used in clinical and preclinical investigations in neuropsychiatric disorders and neurodegenerative diseases (Evrensel and Ceylan, 2016; Gulati et al., 2023; Matheson and Damian Holsinger, 2023; Sanzone et al., 2024; Zhange et al., 2024) and the treatment for recurrent C. difficile (XIFAXAN® or rifaximin approved by FDA).

References

Evrensel A, Ceylan ME. 2016. Fecal Microbiota Transplantation and Its Usage in Neuropsychiatric Disorders. Clin Psychopharmacol Neurosci. 14(3):231-7.

Gulati AS, Nicholson MR, Khoruts A, Kahn SA. 2023. Fecal Microbiota Transplantation Across the Lifespan: Balancing Efficacy, Safety, and Innovation. Am J Gastroenterol. 118(3):435-439. 

Matheson JT, Holsinger RMD. 2023. The Role of Fecal Microbiota Transplantation in the Treatment of Neurodegenerative Diseases: A Review. Int J Mol Sci. 24(2):1001.

Sanzone J, Life M, Reiss D, May D, Hartley B, Spiddle P, Al-Kirwi J, Grigoryan T, Costin J. 2024. Uses of Fecal Microbiota Transplantation in Neurodegenerative Disease: A Scoping Review. Cureus. 16(6):e62265.

Zhang Q, Bi Y, Zhang B, Jiang Q, Mou CK, Lei L, Deng Y, Li Y, Yu J, Liu W and Zhao J 2024. Current landscape of fecal microbiota transplantation in treating depression. Front. Immunol. 15:1416961

Comment 6: Graphical abstract need to change to highlight content of the paper

Response 6: Done as requested (line 72).

Comment 7: add the diagrams for section, 2 (Stress and Injurious Behavior in Domestic Chickens) 3 (Chicken Model in Psychological Disorders) and 4 (4. Gut Microbiota and the Gut-Brain Axis

Response 7: Figures 1-2 have been added as suggested (lines 183 and 306).